# How *PART*s assemble into wholes: Learning the relative composition of images

Melika Ayoughi[1], Samira Abnar[2], Chen Huang[2], Chris Sandino[2], Sayeri Lala[2], Eeshan Gunesh Dhekane[2], Dan Busbridge[2], Shuangfei Zhai[2], Vimal Thilak[2], Josh Susskind[2], Pascal Mettes[1], Paul Groth[1], and Hanlin Goh[2]

[1]University of Amsterdam
[2]Apple

## 1 Abstract

The composition of objects and their parts, along with object-object positional relationships, provides a rich source of information for representation learning. Hence, spatial-aware pretext tasks have been actively explored in self-supervised learning. Existing works commonly start from a grid structure, where the goal of the pretext task involves predicting the absolute position index of patches within a fixed grid. However, grid-based approaches fall short of capturing the fluid and continuous nature of real-world object compositions. We introduce PART, a self-supervised learning approach that leverages continuous relative transformations between off-grid patches to overcome these limitations. By modeling how parts relate to each other in a continuous space, PART learns the *relative composition of images*—an off-grid structural relative positioning that is less tied to absolute appearance and can remain coherent under variations such as partial visibility or stylistic changes. In tasks requiring precise spatial understanding such as object detection and time series prediction, PART outperforms grid-based methods like MAE and DropPos, while maintaining competitive performance on global classification tasks. By breaking free from grid constraints, PART opens up a new trajectory for universal self-supervised pretraining across diverse datatypes—from images to EEG signals—with potential in medical imaging, video, and audio.

## 2 Introduction

Most visual datasets lack the ground truth labels required for supervised learning [1–5]. However, even without relying on expensive labeled data, raw images provide a rich source of information for learning visual representations. Self-supervised learning (SSL) leverages this information by defining pretext tasks [6]. While many pretext tasks focus on global visual invariances to pretrain deep networks [7–15], local spatial structures in images are suitable for precise downstream tasks [16]. Specifically, many works in self-supervised visual learning define local pretext tasks that extract patches from images using a *grid* structure, building on the popularity of grid-based vision transformers. For instance, early well-known SSL methods, often referred to as puzzle solvers [6, 17, 18] shuffle grid-shaped patches from the image and predict the *absolute* position of each patch, as in Jigsaw [19]. The same idea has been applied to pretraining the recent transformer architectures. For example, Masked Auto Encoders [20], MP3 [21], and DropPos [22] mask and shuffle grid-based patches, and regenerate the original unmasked image.

So far, local SSL approaches, such as puzzle solvers and masked image modeling, (i) rely on fixed *grid* structures to patchify images and (ii) predict *absolute* patch positions. Yet real-world objects rarely conform to rigid grid patterns, and absolute position prediction limits the generalizability. In this work, we look beyond these two aspects that current SSL approaches adopt. First, we propose sampling patches freely in location and size. The random *off-grid* sampling enables fine-grained and occlusion-friendly representations. Second, we learn the *relative* relationships between off-grid patches rather than the absolute relations of patches to the image.

Such a relative pretext task allows us to learn visual representations better suited for downstream tasks beyond classification. Pretraining with absolute positions captures only the typical location of a given object in an image, leading to misinterpretations when objects appear in uncommon locations. Pretraining with relative positions enables the model to capture relationships between different objects and the internal composition of object parts, which can be generalized to other objects. Furthermore, while grid-based methods first sample the entire image uniformly and then apply a separate masking step, off-grid sampling inherently results in partial coverage—some regions are sampled while others are left out—thereby integrating masking directly into the sampling process. Off-grid sampling also allows greater control over the location and size of patches—for example, enabling focused sampling on objects, backgrounds, or uniformly across the image. Finally, off-grid sampling is more flexible and can be extended to patches of different scales and aspect ratios to also include odd-shaped objects. Figure 1 shows the difference between our approach and canonical grid-based SSL methods.

We introduce **PA**irwise **R**elative **T**ranslations a

Proceedings of the 7th Northern Lights Deep Learning Conference (NLDL), PMLR 307, 2026.

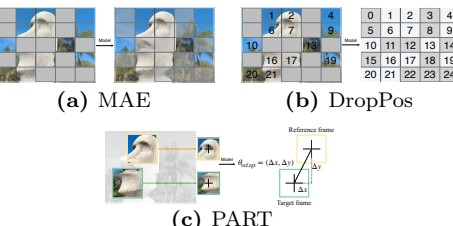

**(a)** MAE          **(b)** DropPos

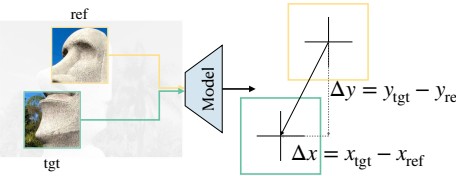

**(c)** PART

**Figure 1. Comparison of PART to other masked image modeling approaches.** (a) generative reconstruction of the masked parts using a fixed grid structure. (b) Masks parts of the image and position embeddings and predicts the absolute patch positions within a pre-defined grid. (c) Unlike grid-based methods, samples patches freely and predicts relative translations between patch pairs, enabling finer spatial understanding.

**Figure 2. Sampling and Objective.** Two random patches are sampled from an image—a yellow reference (ref) patch and a green target (tgt) patch. Given the pixel values of patches, the model predicts the relative translation. The task is to learn the displacement $\Delta x$, $\Delta y$, mapping the position of the reference patch to the target patch. While data augmentations are complementary and applied at the image level to learn global invariances, PART learns local relative representations.

pretraining method that predicts *relative* translations between randomly sampled patches. The pretext objective is set up as a regression task to predict the translation $(\Delta x, \Delta y)$ between each pair of patches (Figure 2). We also introduce a cross-attention architecture that serves as a projection head. We empirically show that PART outperforms baselines in precise tasks such as object detection and 1D EEG signal classification and remains competitive in image classification. We also perform ablation studies on different sampling strategies, projection head architectures, and the number of patch pairs. Code is openly available at https://github.com/Melika-Ayoughi/PART.

# 3    Related Work

Self-supervised learning has emerged as a powerful paradigm for learning meaningful representations from vast amounts of unlabelled visual data, addressing the limitations posed by the scarcity of annotated datasets required for supervised learning [3–6, 23]. By designing pretext tasks, SSL methods exploit the inherent structure of unlabelled images to learn discriminative features without human-annotated labels. These pretext tasks are generally divided into two broad categories: those that operate on the global structure of images and those that focus on local structures within the image. Global methods emphasize invariance to transformations applied to the entire image, while local methods focus on capturing fine-grained spatial structures.

**Global self-supervised learning**   Global pretext tasks have been extensively explored in the literature and focus on leveraging image-wide transformations to learn invariant features. Notable global methods include invariance to rotations [9, 10, 12] and geometric transformations [13–15, 24]. In CIM [25], the model predicts the geometric transformation of a sample with respect to the original image. Other global tasks include colorization [7, 8], denoising [26, 27], and instance discrimination [11, 28]. Building on instance discrimination, contrastive self-supervised methods have emerged, where different views or representations of the same data point are presented to one or two parallel models, with the objective of maximizing agreement between the two views. Common examples include MoCo [29–31] and SimCLR [32, 33] variants. A key concept in contrastive methods is the contrastive loss, which minimizes when the two input images are similar and maximizes when they are dissimilar [34–36]. The fundamental loss function enabling contrastive training for image-based SSL is InfoNCE [37, 38]. Self-labeling through clustering is another SSL approach, with methods such as DeepCluster [39], Self-Classifier [40], CoKe [41] and SwAV [42]. Distillation-based methods avoid the need for negative samples in clustering and contrastive techniques by training a student network to predict the representations of a teacher network [43–47]. Additionally, information-maximization methods focus on maximizing the information conveyed by decorrelated embeddings, eliminating the need for negative samples or asymmetric architectures [48–51]. While these global methods effectively capture high-level patterns, they often miss the rich spatial details present in smaller image regions, which are critical for tasks involving fine-grained image understanding [16]. By focusing on transformations applied to the entire image, they may overlook the nuanced information embedded in local structures.

**Local self-supervised learning**   Real-world images contain a wealth of local patterns, which are essential for understanding the finer aspects of visual content. Local pretext tasks, therefore, shift the focus towards modeling the internal spatial structure of images by extracting small grid-shaped patches from the image and defining pretext tasks on them. An early SSL method for learning local intra-image structures is the Jigsaw puzzle [17, 18], where patches are shuffled, and the model must predict their correct arrangement, commonly referred to as puzzle solvers. Masked image modeling (MIM) emerged as an adaptation of masked language modeling in NLP [52], introducing a new pretext task for self-supervised training [53]. In these methods, part of the input is masked, and the model is tasked with either reconstructing the original input or predicting

the masked-out portion. This can be considered a variant of image inpainting. In Pathak et al. [54], the network is trained to inpaint the contents of a masked image region by understanding the context of the entire image. This approach has also been used to pretrain vision transformers [55], showing improved performance on downstream tasks compared to supervised and contrastive learning baselines.

A popular example of masked image modeling is MAE [20], based on BEiT [56], where a random subset of image patches is masked, and the model reconstructs the entire image in pixel space. Similarly, in I-JEPA [57], the pretext task is to predict the representation of the rest of the image blocks given a single context block, focusing on large-scale models in linear probing. These methods can be categorized as *generative-based* approaches, where the model reconstructs the original input using generative models, such as VAEs [58] or GANs [59].

However, generative-based masked prediction presents challenges, such as longer training times and increased complexity of the reconstruction task [21]. To address these issues, alternative models focus on predicting the *absolute* position of the masked patches instead of pixel reconstruction [21, 22]. In MP3 [21], the corresponding keys to a random set of patches are masked out, whereas, in DropPos [22], the position embeddings of a random portion of the image are masked out. The pretext task in both methods is predicting the exact position of each patch, requiring it to solve the puzzle of determining where each patch originated from. The idea behind these methods originates from works of Doersch et al. and Mundhenk et al. [60, 61] and later on the Jigsaw works [19, 62], where masking is performed by making a puzzle from a part of the image and pretraining a CNN to solve the jigsaw puzzle by predicting the absolute position of each piece. DILEMMA [63] enforces predicting the position of patches that have been artificially misplaced. In Caron et al. [64], the pretext task is the absolute position prediction of a random portion of the image given the input image as a reference. Self-supervised learning has demonstrated significant success across various domains and applications. After the introduction of Vision Transformers (ViT) [55] in 2021, ViTs were quickly adopted for self-supervised learning through methods like BEiT [56], MAE [20], and DINO [45], sparking considerable interest in leveraging these architectures for large-scale unlabeled data in self-supervised learning. While vision transformers often exhibit insensitivity to the order of input tokens [21, 22, 65, 66], suggesting that they tend to model relationships between unordered tokens, the aforementioned models focus explicitly on absolute grid-based position prediction. In contrast, PART is trained on predicting off-grid *relative* translations between random input patches.

**Relative self-supervised learning** The notion of relative information has been applied in self-supervised learning across various tasks and domains. In graph representation learning, Peng et al. [67] proposed predicting the local relative contextual position of one node to another. For single-image depth estimation, Jiang et al. [68] introduced a method for estimating relative depth using motion from video sequences. In the domain of object detection, LIO [69] proposed a self-supervised spatial context learning module that captures the internal structure of objects by predicting the relative positions within the object's extent. LIO localizes the main object in an image and learns the relative positions of other context points with respect to that main object. It operates by defining one main object as a reference, with other points or pixels learning their relative positions to this reference point. Similarly, HASSOD [70] identifies objects in an image through clustering and progressively refines object understanding hierarchically by discovering object parts. Both approaches are tailored to object-centric tasks. In contrast, PART is applicable to both object-centric and object-agnostic data, such as EEG, since it learns relative information by considering any pair of patches as reference and target, allowing it to model relationships between any two patches.

## 4 Method

**Sampling** An overview of our method is shown in Figure 3. Given an image $I \in \mathbb{R}^{H \times W \times C}$, we extract $N$ random patches from the image. $H$ and $W$ are the height and width of the image, and $C$ is the number of channels. With $(x_s, y_s)$ as the coordinates of the top left corner of the patch uniformly sampled across the image and $(x_s + P, y_S + P)$ as the coordinates of the bottom right corner of the patch, respectively. These patches are of shape $P \times P$ and are in random positions of the image. $P$ is the patch size, and $N = \frac{H \times W}{P^2}$ is the number of patches. Now we have $N$ samples of $P \times P \times C$ that can be reshaped into the original image size $\hat{I} \in \mathbb{R}^{H \times W \times C}$. This reshaped image would be akin to a puzzled version of the original image if the random samples were on-grid and with a $P \times P$ shape. In the process of off-grid random sampling, parts of the image are naturally masked out. In addition, some information about each patch's spatial frequency is masked by resizing all samples to the patch size. The pretext task is set up such that the ViT [55], which processes images as sequences of patches, consumes images with incomplete information.

The reshaped patches $\hat{I}$ are then given to the ViT model. In the ViT model, $\hat{I}$ is reshaped into a sequence of patches $I_p \in \mathbb{R}^{N \times (P \times P \times C)}$. A linear projection is then applied to $I_p$, mapping it to $d$ dimensions to get patch embeddings $X \in \mathbb{R}^{N \times d}$. $X$ is given as an input to the transformer blocks

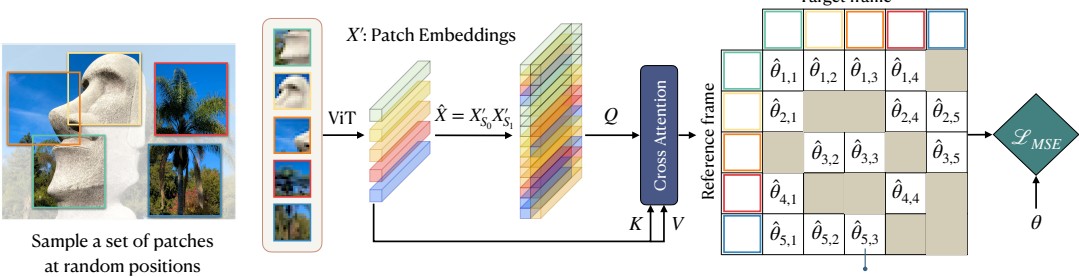

Predicted relative translations: $\hat{\theta}_{ij} = (\hat{\Delta}x, \hat{\Delta}y)_{ij}$

**Figure 3. Illustration of PART on 2D image data:** First, a set of patches is randomly sampled from different positions in the image. This selection is done independently for each image at every iteration. Next, all patches are resized to a uniform size. A ViT model is then used to produce an embedding for each patch. A relative cross-attention encoder then attends to pairs of patches, using a shared weight structure to predict the relative translations between selected pairs. For each pair of reference and target patches $(i, j)$, the cross-attention encoder returns $\hat{\theta}_{ij}$, which represents the relative translation needed to align the reference patch $i$ to the target patch $j$.

without the position embeddings. The ViT model returns the learned patch embeddings $X' \in \mathbb{R}^{N \times d}$.

**Objective** Before discussing how the patch embeddings $X'$ are utilized in the objective function, it is important to first introduce the different components of the objective function and how it is parameterized. This requires us to revisit the off-grid sampling process and how it produces the target values for the objective function. First, a pair of patches (reference, target) are sampled from the image at random positions with $(x_{\text{ref}}, y_{\text{ref}})$ and $(x_{\text{tgt}}, y_{\text{tgt}})$ representing the center pixel coordinates of the two patches in image space. Both patches are then resized to a uniform patch size $P$, masking their original position in the image space, as well as their pixel content. The goal is to learn the underlying translation between any pair of patches. This translation transforms the reference coordinate patch into the target coordinate patch, considering the width $w_{\text{ref}}$ and height $h_{\text{ref}}$ of the reference patch. The task is to predict

$$\theta_{\text{ref},\text{tgt}} = \begin{bmatrix} \Delta x \\ \Delta y \end{bmatrix}_{\text{ref},\text{tgt}} = \begin{bmatrix} (x_{\text{tgt}} - x_{\text{ref}})/w_{\text{ref}} \\ (y_{\text{tgt}} - y_{\text{ref}})/h_{\text{ref}} \end{bmatrix}_{\text{ref},\text{tgt}}$$

with $\Delta x$ and $\Delta y$ capturing *relative position*. In simple terms, the goal is to move the reference frame so that it translates into the target frame. In this context, when referring to a "frame", we specify the bounding box itself rather than the actual pixel contents in the bounding box i.e. the patch (Figure 2). In contrast to augmentations that are applied to entire images (e.g., rotations, scaling) to enforce global invariance, PART focuses on learning spatial relationships within the image, capturing the relative geometry between regions.

The emphasis on predicting the relative translation is key because the pixel space information is lost after resizing patches to a uniform size. After resizing, the model no longer possesses details about the original image space and needs to learn to be robust to different image resolutions. The two terms we seek to predict are the translation in $x$ normalized by the width of the reference patch $w_{\text{ref}}$ and

the translation in $y$ normalized by the height of the reference patch $h_{\text{ref}}$. In this case, both $w_{\text{ref}}$ and $h_{\text{ref}}$ are equal to patch size $P$ due to resizing.

**Relative encoder architecture** The ViT model outputs a per-patch embedding $X' \in \mathbb{R}^{N \times d}$. The relative encoder maps the per-patch embeddings to the relative translations between a random number of patch pairs ($\#pairs$), resulting in $\hat{\theta} \in \mathbb{R}^{2 \times \#pairs}$. The two outputs per patch pair are the relative positions between the reference and target patches. Given $X'$, this module selects random index pairs ($\#pairs$) of patches $S \in \mathbb{N}^{2 \times \#pairs}$ with $S_0$ as the index of the reference patch and $S_1$ as the index of the target patch. The embeddings of reference patches $S_0$ and $S_1$ are then concatenated: $\hat{X} = \texttt{concat}(X'_{S_0}, X'_{S_1})$. $\hat{X}$ goes through a linear projection to convert from $\mathbb{R}^{\#pairs \times 2 \times d}$ to $\mathbb{R}^{\#pairs \times d}$. $\hat{X}$ is fed into a cross-attention module [71] as the query, and $X'$ is fed as both the key and the value. $d$ is the dimensionality of the keys/queries: $\hat{\theta} = \texttt{softmax}\left(\frac{\hat{X}X'^{\top}}{\sqrt{d}}\right) X'$. The cross-attention module allows for information dissemination between all patch embeddings and enables the model to focus on predicting the relative translation only for a subset $S$ of patch pairs. This imposes further masking of information given to the model. We discuss the pros and cons of this design choice in depth in the ablations section. $\theta$ is only calculated for the subset $S$ of patch pairs. The model is trained with a mean squared error loss between the predicted relative translations $\hat{\theta}$ and the ground-truth relative translations $\theta$:
$\mathcal{L}_{MSE} = \frac{1}{2 \times \#pairs} \sum_{k=1}^{2} \sum_{m=1}^{\#pairs} \left( \theta_{mk} - \hat{\theta}_{mk} \right)^2$.

**Training setup** Once the model is pretrained, we tune the network end-to-end using labeled data in a supervised setup. Following the standard ViT setup, we eliminate the relative encoder and substitute it with a linear classification or detection head after the [CLS] token, which aggregates global input information. Unlike pretraining, we incorporate randomly initialized learnable position embeddings and apply

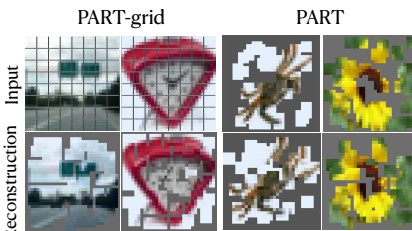

**Figure 4. Target image reconstruction** given predicted relative translations in PART vs. grid sampling.

fixed grid sampling instead of random sampling and masking.

## 5  Results

Having introduced the method, we now analyze the properties learned by PART in practice. We begin with qualitative capabilities that arise from off-grid sampling and relative patch prediction, and then evaluate PART quantitatively against established baselines.

### 5.1  Capabilities of PART

In this section, we start by emphasizing the potential capabilities that emerge from adopting relative off-grid sampling in SSL frameworks, followed by comparisons to other methods. These capabilities highlight the unique advantages and transformative possibilities that such a paradigm shift offers, namely, (i) off-grid reconstruction, (ii) extension to other aspect ratios and scales, (iii) patch uncertainty, and (iv) symmetry. For reproducibility, implementation, hyperparameters and the choice of architecture is explained in depth in the Appendix in section 8.1.

**(i) Off-grid reconstruction**   Unlike grid-based approaches, PART can reconstruct the original image from off-grid patches. This is especially valuable in domains like Satellite and LiDAR imaging, where overlapping patches from different images must be reassembled. These patches need not come from a single image—for instance, the model can compose a new face by arranging parts of different faces. Figure 4 illustrates this: input patches (top row) and the predictions under different sampling strategies (bottom row). In PART, the ground truth visualization consists of a subset of the patches, thus providing a masked input to the model. Images are generated by fixing one random reference patch and positioning all other patches relative to it. In grid sampling, the ground truth positions reconstruct the full image, since patches cover the entire image. The model's predictions nearly match the ground truth, even in fine details, having learned general scene structure (e.g., sky above, road below, clock's triangular form). Some details are missing, such as clock hands and numbers, and mono-color patches are harder to place since the model sees only pixel content. PART's ability to reconstruct from off-grid patches highlights its grasp of underlying image structure.

**(ii) Extension to multiple aspect ratios and scales**   The combination of off-grid patch sampling and relative position prediction as a pretext task, offers the opportunity to reimagine patch sampling in the vision transformers. If the method can predict relative positions of equal-sized square patches (1:1 aspect ratio), why not also between patches of varying aspect ratios and scales? To test this, we run a proof-of-concept experiment extending the objective to include $\Delta w$ and $\Delta h$ for relative width and height:

$$\theta_{\mathrm{ref,tgt}} = \begin{bmatrix} \Delta x \\ \Delta y \\ \Delta w \\ \Delta h \end{bmatrix}_{\mathrm{ref,tgt}} = \begin{bmatrix} (x_{\mathrm{tgt}} - x_{\mathrm{ref}})/w_{\mathrm{ref}} \\ (y_{\mathrm{tgt}} - y_{\mathrm{ref}})/h_{\mathrm{ref}} \\ w_{\mathrm{tgt}}/w_{\mathrm{ref}} \\ h_{\mathrm{tgt}}/h_{\mathrm{ref}} \end{bmatrix}_{\mathrm{ref,tgt}}$$

We modify the sampling to include bottom-right coordinates $(x_e, y_e)$ of the patches, then resizing patches to the ViT patch size $P \times P$. Patch width and height are constrained between half and twice the ViT patch size to ensure meaningful content. The model is pretrained with the new objective for 100 epochs, and results on ImageNet classification and COCO detection are reported in Table 1. Extending grid-based methods like MAE to multiple aspect ratios and patch sizes is non-trivial, requiring more advanced positional embeddings and a decoder that can upsample multi-scale representations.

**Table 1. Extending PART to patches of different aspect ratios & scales:** comparison of PART and its extension on COCO Object Detection and ImageNet classification without extra hyperparameter tuning. This proof-of-concept experiment motivates a new avenue for further research in relative off-grid pretext tasks.

|  | COCO OD | | | INet Class. |
|---|---|---|---|---|
|  | $\mathrm{AP}^b$ | $\mathrm{AP}^b_{50}$ | $\mathrm{AP}^b_{75}$ | Accuracy |
| PART | 42.4 | 62.5 | 46.8 | 82.7 |
| PART + aspect ratio + scale | 42.0 | 61.8 | 46.3 | 82.6 |

In practice, the extended model is trained for the same number of epochs and with the same hyperparameters as the base model, without additional tuning. The slight performance drop is likely due to the increased parameter and objective complexity introduced by the additional $\Delta h$ and $\Delta w$ terms. We observed in our experiments that the convergence of scale and aspect ratio was fast, while delaying the convergence of $\Delta x$ and $\Delta y$ compared to before.

**(iii) Patch uncertainty**   Another capability of PART is estimating patch uncertainty by checking whether different reference patches agree on the relative position of a target patch. Our model predicts the relative translation both when patch $i$ serves as a reference and target patch. In Figure 5, one target patch is positioned relative to all reference patches. If patches are clustered, the model is more certain of that patch. Such uncertainty estimation is valuable in applications like semantic segmentation for autonomous vehicles or tumor detection, where it

helps distinguish common from anomalous scenarios.

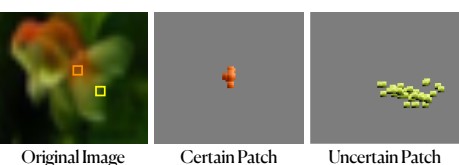

Original Image    Certain Patch    Uncertain Patch

**Figure 5. Patch uncertainty**: A single target patch is positioned relative to all reference patches. Left: two target patches (orange, yellow). Middle/right: model uncertainty for orange and yellow, respectively. The model is more certain about the unique orange patch, while the yellow patch resembles other regions, making its position harder to predict. Easy patches (orange) are consistently placed at the same location, showing that some patches are more confidently localized than others.

**(iv) Symmetry** When observing a lip, one expects a nose above it; seeing a nose suggests a lip below. This illustrates symmetry. In this experiment, we demonstrate that PART learns and represents these symmetrical relationships. Figure 6 compares the model's prediction matrix with the ground truth, showing strong alignment along both $x$ and $y$ axes. The key property that emerges from this figure is negative symmetry: if patch $P_i$ predicts $(\Delta x, \Delta y)$ relative to $P_j$, then $P_j$ predicts $(-\Delta x, -\Delta y)$ relative to $P_i$. Despite heavy masking and lack of global patch information, the model positions patches correctly relative to each other. Learning the negative symmetry results in consistent relative positioning of object parts, which we expect benefits localization and fine-grained understanding. When samples are scarce, our approach allows for learning from fewer variations due to these built-in symmetries.

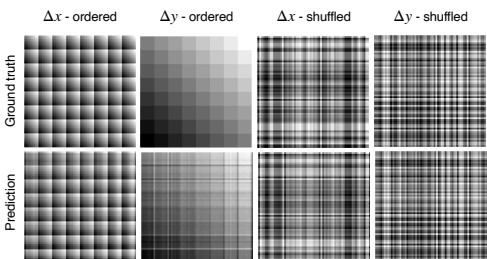

**Figure 6. Negative symmetry in output prediction matrices** for $x$ and $y$ translations with ordered vs. shuffled patch indices, showing that PART positions each patch-pair correctly relative to each other. Ordered matrices sort patches from top-left to bottom-right. Each $N \times N$ element $(i, j)$ gives the relative translation from reference patch $i$ to target $j$. Bright colors indicate positive values, dark negative, and gray zero; color intensity reflects magnitude.

## 5.2 Comparison to grid-based

So far, we have qualitatively demonstrated that PART learns to reconstruct the target image using the relative positions of off-grid patches. It also learns the negative symmetry between pairs of patches. These capabilities arise because PART has learned the structure of input images and how they relate to each other both locally and globally within the image. Here, we focus on quantitative results and compare PART to other grid-based methods that provide a fair comparison on precise local tasks, such as object detection and time-series prediction.

**Object detection** In Table 2, we compare PART with MAE [20], MP3 [21] and DropPos [22] grid-based pretraining methods in the downstream object detection performance. The results demonstrate that PART's off-grid sampling with overlapping patches and relative patch position prediction improve detection accuracy, particularly for fine-grained local tasks where precise spatial understanding is critical, outperforming methods like MAE, MP3, and DropPos, which rely on fixed grid-based sampling. Notably, while MAE and DropPos use positional embeddings, MP3 and PART do not. PART achieves performance comparable to DropPos, despite DropPos employing additional losses (attentive reconstruction and position smoothing). Without these auxiliary losses, DropPos's detection performance drops by roughly 2% (Tables 3 & 4 in [22]), highlighting the strength of PART's objective.

**Table 2. Object detection comparison.** Our method outperforms state-of-the-art grid-based baselines on COCO detection while relying on the same backbone. † means our implementation, ♯ means the result is borrowed from [22].

|  | $AP^b$ | $AP^b_{50}$ | $AP^b_{75}$ |
|---|---|---|---|
| *Grid-based* | | | |
| MAE [20]♯ | 40.1 | 60.5 | 44.1 |
| MP3 [21]† | 41.8 | 61.4 | 46.0 |
| DropPos [22] | 42.1 | 62.0 | 46.4 |
| *Relative off-grid* | | | |
| **PART** | **42.4** | **62.5** | **46.8** |

**Time-series prediction** PART can also be applied to 1D data. Here we take 1D time series prediction as an example. For 1D data, PART predicts relative time shifts ($\Delta t$) between randomly-sampled, unequally-spaced windows (the 1D equivalent of off-grid patches) from longer sequences. We validate this approach by pretraining a 1D ViT on biosignals from the PhysioNet 2018 "You Snooze You Win" Challenge Dataset [72]. As shown in Table 3, our method achieves at least a 2% improvement in sleep staging classification performance compared to both supervised and self-supervised baselines. This task particularly benefits from PART's capabilities, as accurate sleep staging requires both precise local representations and global understanding of each stage's position within the complete sequence. Additionally, as shown in the Appendix 8.2, PART demonstrates superior sample efficiency by effectively learning the structure of 1D EEG signals, although limited training data is available.

**Table 3. Sleep stage classification** accuracy represented using Cohen's Kappa. PT, FT = number of pretraining, finetuning epochs. †= our implementation.

|  | PT | FT | Cohen's Kappa |
|---|---|---|---|
| *Supervised* |  |  |  |
| Supervised w/ Pos Embed† | 0 | 100 | 0.531 |
| *Grid-based* |  |  |  |
| MP3 [21]† | 1000 | 100 | 0.553 |
| DropPos [22]† | 1000 | 100 | 0.582 |
| MAE [20]† | 1000 | 100 | 0.595 |
| *Relative off-grid* |  |  |  |
| PART | 1000 | 100 | **0.616** |

## 5.3 Ablations

**Sampling strategies** An essential component of our method is the patch sampling process. Besides random sampling, we ablate on on-grid sampling similar to MP3 and DropPos (Figure 7). In the grid sampling, all patches are arranged in a grid form, with a fixed size at fixed positions. PART-grid has a similar patch sampling to MP3 but with a relative objective function. The results in Table 4 show that

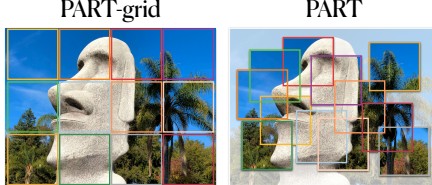

**Figure 7.** PART adopts a random sampling strategy. Grid sampling (PART-grid) is performed as an ablation.

random continuous sampling improves performance in different tasks and domains compared to PART-grid while introducing more masking. Looking beyond the grid improves downstream performance.

|  | COCO | CIFAR-100 | IN-1K | Time-series |
|---|---|---|---|---|
| PART-grid | 41.4 | 82.1 | 82.43 | 0.500 |
| PART | **42.4** | **83.0** | **82.7** | **0.616** |

**Table 4. Apples-to-apples comparison** for on-grid versus off-grid patch sampling with relative position prediction objective. For detection (COCO), classification (CIFAR-100, IN-1K), and time-series prediction, off-grid patch sampling performs better since it can capture precise local information better.

**Impact of relative encoder** Besides the cross-attention relative encoder in the method, we perform an ablation study on two other ways to learn this mapping. The most straightforward approach is a fully connected MLP that receives all patches concatenated as an input and predicts the translation for any two patches. So, given $N$ patches with $d$ dimensions, the relative encoder would have $N * d * N^2 * 2$ parameters. Although the weights are not shared in this approach, such as in the cross-attention head, the relative encoder can access all patch embeddings. This helps the model to converge faster because it can use extra information from other patches. However, the classification head will replace the relative

encoder during finetuning. The time spent on training the fully connected MLP can be spent on training better representations instead.

**Table 5. Ablation on different relative encoders** for CIFAR-100 pretrained for 1000 epochs. The cross-attention is preferred over standard feed-forward layers.

|  | Error ↓ |  |  | Accuracy ↑ |
|---|---|---|---|---|
|  | $x$ | $y$ | Euclidean |  |
| MLP | 3.18 | 2.02 | 1.68 | 82.38 |
| Pairwise MLP | 2.84 | 1.76 | 1.59 | 82.52 |
| Cross-attention | **1.14** | **0.77** | **0.81** | **83.00** |

We propose an alternative relative encoder that compensates for the high parameter count in the fully connected MLP approach through weight sharing, which we term a pairwise MLP. The pairwise MLP receives two concatenated patches as input and predicts their relative translation. Although this approach uses only $2 * d * 2$ parameters, the relative encoder cannot access all the patches, thus predicting the translations solely based on the content of these two patches. Table 5 shows the results for different relative encoders. The results suggest that the cross-attention head (83.00%) outperforms pairwise MLP (82.52%) and MLP (82.38%). MLP is computationally more expensive than pairwise MLP and cross-attention.

**Table 6. ImageNet-1k classification with ViT-B.** PART is comparable to other grid-based methods. Pos Embed = using position embedding. †= our implementation, ♯= borrowed from [20], ∗= borrowed from [21].

|  | Pos Embed | PT | FT | Accuracy |
|---|---|---|---|---|
| *Supervised* |  |  |  |  |
| Labelled baseline* | ✓ | 0 | 300 | 81.8 |
| Labelled baseline* |  | 0 | 300 | 79.1 |
| *Contrastive* |  |  |  |  |
| MoCo v3 [66]♯ | ✓ | 300 | 150 | 83.2 |
| DINO [45]♯ | ✓ | 300 | 300 | 82.8 |
| BEiT [56]♯ | ✓ | 800 | 100 | 83.2 |
| CIM [25] | ✓ | 300 | 100 | 83.1 |
| *Grid-based* |  |  |  |  |
| MAE [20]* | ✓ | 150 | 150 | 82.7 |
| MAE [20]* | ✓ | 1600 | 100 | 83.6 |
| MP3 [21]† | ✓ | 400 | 300 | 82.6 |
| MP3 [21] |  | 100 | 300 | 81.9 |
| DropPos [22] | ✓ | 200 | 100 | 83.0 |
| *Relative off-grid* |  |  |  |  |
| PART |  | 400 | 300 | 82.7 |

***Does PART come at the cost of image classification?*** In Table 6, we compare PART with supervised and state-of-the-art SSL alternatives on the ImageNet-1K [73] classification benchmark. Our method outperforms the supervised results as well as MP3 [21] and shows competitive performance with respect to DropPos [20] and MAE [20]. Note the latter methods employ position embedding during pretraining. DropPos employs extra position

**Table 7. CIFAR-100 ViT-S**. PART is comparable to other grid-based methods. ♯= borrowed from [21].

| | Pos Embed | PT | Accuracy |
|---|:---:|:---:|:---:|
| *Supervised* | | | |
| Labelled baseline♯ | ✓ | 0 | 73.6 |
| Labelled baseline♯ | | 0 | 64.6 |
| *Contrastive* | | | |
| MoCo v3 [66] ♯ | ✓ | 2000 | 83.3 |
| *Grid-based* | | | |
| MAE [20]♯ | ✓ | 2000 | 84.5 |
| MP3 [21] | ✓ | 2000 | 84.0 |
| MP3 [21] | | 2000 | 82.6 |
| *Relative off-grid* | | | |
| PART | | 1000 | 83.0 |

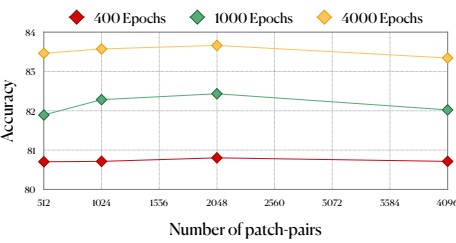

**Figure 8. #patch_pairs ablation** CIFAR-100.

smoothing and attentive reconstruction techniques that could further accelerate training.

We compare PART with supervised and self-supervised alternatives on the CIFAR100 [74] classification benchmark in Table 7. PART consistently outperforms the supervised baselines with and without position embeddings, although it does not use any position embeddings. With only 1000 pretraining epochs, PART outperforms the MP3 [21] baseline with 2000 epochs of pretraining.

**Number of patch pairs** As explained in 4, a subset $S$ is randomly chosen from the patch embeddings. #*pairs* is the parameter that determines the length of $S$. We study the effect of #*pairs* in Figure 8 after 400, 1000, and 4000 epochs of pretraining. We observe that curves follow similar patterns for different epochs of pretraining, while more pretraining epochs result in higher accuracy. We also observe a trade-off in #*pairs*. Higher #*pairs* means the model sees more patch information but must also predict the relative translations for more contradicting patch pairs. Whereas smaller #*pairs* means the model has access to less information, thus overfitting on the task leading to less general representations. There is a sweet spot with 2048 patch pairs, where enough global patch information is given to the model, and the training task is neither easy nor difficult.

## 6 Conclusion

The composition of objects and their parts, along with their relative positions, offers rich information for representation learning. We introduced PART, a pretraining method that predicts continuous relative transformations between random off-grid patches, learning the relative composition of images that generalize beyond occlusions and deformations. We demonstrated PART's capabilities—off-grid reconstruction, flexible patch forms, patch uncertainty, and symmetry—and how these support the quantitative results. On tasks requiring precise spatial understanding, such as object detection and time-series prediction, PART outperforms grid-based methods like MAE and DropPos, while remaining competitive on global classification. Our experiments show PART's applicability across data types, domains, and tasks, with potential for further extensions discussed in the next section.

## 7 Discussion & Future Work

So far, we demonstrated the capabilities of PART as well as its applicability on multiple data types (1D & 2D), domains (medical & every-day) and tasks (classification & detection). Here, we discuss potential benefits and future directions in depth.

**Complementary to contrastive learning:** PART provides fine-grained local representations, making it a complement to contrastive methods. Combined, they can capture both local and global patterns by uniting PART's off-grid position prediction with contrastive learning's view augmentations.

**Hierarchical multi-scale learning:** PART's sampling strategy raises questions on whether patches should be sampled randomly or focus on objects or background depending on the downstream task. Extending to multiple scales and aspect ratios could enable hierarchical multi-resolution representations, where objects and their parts at different scales are accurately captured.

**Modeling rotations:** For example, seeing a lip suggests a nose above it—but if the lip is rotated, the expectation is a rotated nose. A key question is how PART can be generalized for rotation equivariance.

**Extension to other tasks:** Relative position prediction strengthens spatial reasoning in continuous space, benefiting tasks that demand fine-grained spatial understanding such as scene graph generation, spatial relation prediction, and 3D reconstruction.

**Universal pretraining across diverse data types and domains:** PART can be extended to audio spectrograms, videos, and sensor data by adding a temporal constraint, learning relative spatio-temporal relationships. Off-grid sampling with overlapping, variable-sized patches enables flexible representations that capture real-world structures. This makes PART useful for reconstruction in satellite and LiDAR imaging, as well as for data-scarce, high-precision domains like medical imaging.

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

# 8 Appendix

## 8.1 Implementation details

For both CIFAR-100 and ImageNet-1k, our pretraining and finetuning configurations closely align with the pipeline outlined in MP3 [21], which itself builds upon the foundational work of [75]. We adopt the MP3 codebase as our starting point for implementation. During the finetuning phase, we adhere strictly to the supervised training protocols recommended in DeiT [75], ensuring consistency with established practices. Detailed descriptions of the implementation specifics for each task are provided below:

**Object detection** We evaluate the transfer learning capacity of our method on the COCO Dataset. We perform self-supervised pretraining on the ImageNet-1K [73] with a resolution of 224×224 using ViT-B [75] as the backbone. The model is pretrained for 200 epochs with a learning rate of 0.0005, a batch size of 1024 with 4802 patch pairs on 8 GPUs. We perform end-to-end finetuning on COCO [76] for object detection. Specifically, Mask R-CNN [77] is finetuned with 1× schedule (12 epochs) and 1024×1024 resolution. We use the configuration of ViTDet [78]

and take ViTB/16 [55] as the backbone. $AP^b$ is reported as the main performance metric for object detection. We additionally report $AP^b_{50}$ and $AP^b_{75}$ that correspond to Average Precision at IoU thresholds of 0.5 and 0.75, respectively, following standard COCO evaluation protocols.

**1D time series classification** PhysioNet 2018 "You Snooze You Win" Challenge Dataset [72] contains multi-channel biosignals that are continuously monitored overnight during sleep studies conducted at Massachusetts General Hospital. The task is to predict one of five sleep stages (Wake, Non-REM1, Non-REM2, Non-REM3, REM) given a 30-second window of data containing six channels of scalp electroencephalography (EEG). The EEG data is bandpass filtered with cutoffs 0.1-30 Hz and then re-sampled to have a 100 Hz sampling rate. Thirty-second windows are instance normalized as an additional pre-processing step before being tokenized by a linear layer. Forty 1-second patches are then randomly sampled. We use recordings from 1,653 subjects across the entire dataset for all pretraining strategies. For finetuning, we down-sample the dataset to 10 subjects to simulate a low-labeled data regime. For testing, the finetuned models are evaluated on recordings from 200 subjects, which are held out from both pretraining and finetuning stages. We randomly mask out 20% of the position embeddings for this experiment. All experiments use a 1D ViT backbone with 12.9M parameters and the input to the model is the 1D data for both pretraining and finetuning.

**Image classification** We perform self-supervised pretraining on CIFAR100 [74] and ImageNet-1K [73] with a resolution of 32x32 and 224×224 respectively. Following Zhai et al. [21], we use ViT-S as the backbone of CIFAR100 and ViT-B [75] as the backbone of ImageNet-1K. During pretraining, we perform a hyperparameter search on learning rates $\{0.0005, 0.001, 0.01\}$, and the number of pairs $\{512, 1024, 2048, 4096\}$ and choose the best result for each experiment. We perform 400 epochs of finetuning on CIFAR100 and 300 epochs of finetuning on ImageNet-1K with a learning rate of $5e^{-4}$. We report accuracy, $\ell_2$ error, and the mean squared error in $x$ and $y$ dimensions.

**Choice of Architecture** Due to computational limitations, we employ vision transformer architectures tailored to the specific datasets and tasks. For pretraining on the CIFAR-100 dataset, we utilize the smaller ViT-S model, while for the ImageNet dataset, we adopt the larger ViT-B model to accommodate its greater complexity and scale. For one-dimensional (1D) data, we implement a specialized 1D variant of the vision transformer to ensure

optimal performance and compatibility with the data structure.

## 8.2 Sample efficiency

To evaluate sample efficiency, we varied the number of subjects used for fine-tuning in the EEG sleep staging task from 10 to 657, and repeated 5 times with different random seeds and sample subsets. As shown below, PART consistently outperforms baselines across all label fractions with fewer fine-tuning samples.

| # Fine-tuning Samples | MP3 | DropPos | MAE | PART |
|---|---|---|---|---|
| 10 | 0.56 | 0.58 | 0.62 | **0.64** |
| 50 | 0.63 | 0.64 | 0.65 | **0.67** |
| 100 | 0.65 | 0.66 | 0.67 | **0.68** |
| 657 | 0.68 | 0.68 | 0.68 | **0.70** |

