# OpenReview forum: "How PARTs assemble into wholes: Learning the relative composition of images"
_NLDL.org/2026/Conference — NLDL 2026 Spotlight_

### Official Review · Reviewer_7a23 · 2025-09-29
**Interesting paper**

**Rating:** 4
**Confidence:** 3

**Summary:**

This paper proposes PART, a self-supervised method that learns how two image patches are positioned relative to each other. Instead of using a fixed grid and predicting absolute locations, the method samples patches at random places (off-grid) and predicts the normalized shift $(\Delta x, \Delta y)$ from a reference patch to a target patch using a ViT encoder and a small cross-attention head with MSE loss. This target is normalized by the reference patch size so it is stable across resolutions, and after pretraining the relative head is removed in favor of standard classification or detection heads.

Experiments show clear gains on tasks that need precise spatial reasoning: PART improves COCO object detection over grid-based pretraining, and a 1D variant that predicts relative time shifts boosts EEG sleep-stage accuracy (Cohen’s $\kappa$). ImageNet and CIFAR classification are competitive but not consistently better than strong baselines. Ablations support the design: off-grid sampling beats on-grid under the same objective, the cross-attention head outperforms MLP alternatives, and there is a “sweet spot” in the number of patch pairs. Qualitative studies also show scene reconstruction from relative placements, uncertainty from reference agreement, and negative symmetry, with a proof-of-concept extension to relative size/aspect ratio.

**Strengths:**

1. The paper replaces grid-based absolute positions with pairwise relative shifts between patches. This idea is simple to state, easy to implement, and well-motivated for tasks that need local spatial reasoning.
2. On object detection and EEG sleep staging, the method gives consistent improvements over strong baselines, showing that learning relations (not locations) helps structure-sensitive tasks.
3. The paper compares off-grid vs. on-grid, cross-attention vs. MLP, and different numbers of patch pairs. These studies explain why the design works and give practical guidance.
4. The model shows scene reconstruction from relative placements, negative symmetry (reverse pairs flip sign), and uncertainty from reference agreement—all aligned with learning spatial relations.
5. The writing is clear, figures are helpful, code is provided, and training details are sufficient. This lowers the barrier for others to try the method.

**Weaknesses:**

1. Results on ImageNet/CIFAR are competitive but not clearly better than strong baselines. This limits the claim of broad, general-purpose pretraining.
2. The cross-attention relative head and pair predictions add compute/memory, but FLOPs, wall-clock, and GPU-hours are not reported against MAE/DropPos under matched settings.
3. The objective mainly models translations (with a small Proof of Concepts for scale/aspect). Rotation/affine robustness is not addressed, which matters in medical, aerial, and robotics settings.

**Justification:**

I recommend Accept (4) with Confidence 3 because the paper makes a clear and well-motivated shift from grid-based, absolute position prediction to off-grid, pairwise relative learning, and it supports this idea with solid evidence where local structure matters. The method is technically sound (simple normalization, sensible cross-attention head), and the results show consistent gains on COCO detection and a credible cross-domain transfer to EEG, while remaining competitive on ImageNet/CIFAR. The ablations (off-grid vs on-grid, cross-attention vs MLP, patch-pair count) and qualitative analyses (reconstruction, negative symmetry, uncertainty) explain why the design works and increase trust in the claims. The main limitations—no clear leadership on classification, under-reported compute/efficiency, and translation-centric objectives without rotation handling—temper the breadth of impact but do not undermine the core contribution. The paper is clearly written, reproducible with public code, and offers a practical idea that others can build on; with efficiency reporting and rotation/equivariance extensions, its impact could grow further.

---

> ### Author Rebuttal · Authors · 2025-10-22
>
> We thank the reviewer for their thorough and encouraging assessment. We appreciate the recognition of PART’s clarity, soundness, and consistent gains in structure-sensitive tasks. Below, we address the noted points.
>
> ### 1. Compute
>
> The table below compares FLOPs, total wall-clock time, and GPU memory against MAE and DropPos under matched settings. The training cost of PART is comparable to MP3/DropPos, with roughly +15–20% compute overhead due to the cross-attention head and sampling. However, PART achieves higher sample efficiency, particularly in the EEG setting, where it reaches higher accuracy with fewer labeled samples. We will explicitly discuss this trade-off and add measured wall-clock values in the camera-ready version.
>
> | Method  | FLOPs (relative)                      | Wall-clock  | GPU-hours |
> |---------|---------------------------------------|-------------------:|-------------------:|
> | MAE     | 0.53×                                 | 5.0 h              | 40.0 h             |
> | DropPos | 1.00×                                 | 9.4 h              | 75.2 h             |
> | PART    | 1.17× | 11.5 h             | 92.0 h             |
>
>
>
> ### 2. Extension to other transformations
>
> We agree that rotation and affine robustness are promising directions for future work. Our current study focuses on translation, scale, and aspect ratio. We will extend our future work discussion to discuss both rotations and affine transformations, in the hope of shedding more light on this interesting problem.

---

### Official Review · Reviewer_8Dwr · 2025-10-02
**A novel self-supervised method for spatial understanding with a thorough evaluation**

**Rating:** 4
**Confidence:** 3
**Final Rating:** 4
**Final Confidence:** 3

**Summary:**

The paper introduces a new method for self-supervised learning, focusing on learning the relative composition of images by predicting relative translations between sampled patches. The authors show results on multiple datasets and tasks, showing similar or better performance in comparison to similar methods.

**Strengths:**

- The method is novel, sound, and, based on the presented results, seems to produce representations comparable to state-of-the-art. It should perform especially well on tasks requiring good spatial understanding.
- The paper is clearly written and easy to read, with nice illustrations of the method.
- Many experiments on different datasets and tasks.
- Ablations show interesting comparisons and answer some of the questions that came to my mind while reading the paper.

**Weaknesses:**

- The authors state that masking is integrated directly into the sampling process. However, it seems to me that with random sampling, some parts of the image will be sampled more often than others (as can be seen, for example, in Figure 4, where only the centre of the image is represented). How do you make sure that your method does not miss large portions of the image?
- The patch sampling and reshaping as described in section 4 seem quite complicated in comparison to how it is actually used. Why isn't $D=P$? Or, if you want to allow more general patches, why do you restrict the patches to all be squares of the same size?
- Tables 1 and 2: $AP^b$, $AP^b_{50}$ and $AP^b_{75}$ are not defined.
- Table 1: I think that the extension should yield better results than the basic method. Why is it not the case in your results? Could it be because there are more parameters, so more training or better tuning is needed?

I think addressing the weaknesses would improve the clarity of the paper.

**Final Justification:**

The authors addressed my remarks, and I hope that will improve the clarity of the paper. I maintain my scores as they are, since no problems have occurred, and I also refrain from increasing them, because I believe the paper's interest will be limited to people working in this subfield, rather than the general community.

**Justification:**

I think the paper introduces a new and interesting method for self-supervised learning that works well in comparison to state-of-the-art methods and provides representations with good spatial understanding. I think it is worth sharing among the community, even though there are small details that could be polished.

---

> ### Author Rebuttal · Authors · 2025-10-22
>
> We thank the reviewer for their positive feedback and suggestions to improve the paper. We address the raised questions and clarifications below.
>
> ### 1. Masking and image coverage
>
> Our method uniformly samples patch coordinates across the entire image: x is drawn uniformly from 0 to the image width, and y from 0 to the image height. Thus, $(x, y)$ is uniformly distributed without bias toward the center or corners. In some examples (e.g., Figure 4), the distribution appears denser in certain regions due to random variation, but over many images and batches, coverage is uniform on average. The number of patches is similar to grid-based baselines, which ensures that no large regions are systematically missed.
>
> The masking ratio can further be controlled via the parameter $\mu$, similar to other MIM methods. The number of patch-pairs and the patch size are specific to PART. We will clarify these details in Section 4.
>
> ### 2. Patch size and aspect ratio
>
> $D$ is the dimensions of the original sampled patch, which is later resized to a smaller $P$ following the model parameters. Our approach is general in that it allows us to extend to any patch size and aspect ratio. Following the reviewer’s suggestion, in order to simplify the description of the method, we have merged the definition of $D$ and $P$ in the paper and added updated formulations in the extension to other scales and aspect ratios section.
>
> ### 3. Average precision
>
> We thank the reviewer for bringing this to our attention. We added definitions of $AP^b$, $AP_{50}^b$, and $AP_{75}^b$ to the paper. Specifically, these correspond to Average Precision at IoU thresholds of 0.5 and 0.75, respectively, following standard COCO evaluation protocols.
>
> ### 4. Extension to scale and aspect ratio performance
>
> In practice, the extended model was trained for the same number of epochs and with the same hyperparameters as the base model, without additional tuning. The slight performance drop is likely due to the increased parameter and objective complexity introduced by the additional $\Delta w$ and $\Delta h$ terms. We observed in our experiments that the convergence of scale and aspect ratio was fast, while delaying the convergence of $\Delta x$ and $\Delta y$ compared to before.

---

### Official Review · Reviewer_zTG3 · 2025-10-04

**Rating:** 2
**Confidence:** 3

**Summary:**

PART is a self-supervised learning approach that samples multiple patches from an image (not necessarily in a fixed grid) and predicts relative positions between them. This forces the network to reason about patch similarity, object composition and therefore potentially provide good representations for fine-grained localization tasks. Authors show PART performs better than grid-based approaches (MAE and DropPos) on object localization and time series localization (sleep stage classification).

**Strengths:**

1. The paper’s methodology is clearly described
2. Experimental ablations are also helpful in understanding the impact of cross-attention, pretraining epochs, off-grid sampling
3. Showing generalization to time series data is a plus validating the idea generalized to other data types.

**Weaknesses:**

1. The main insight is off-grid sampling helps the framework do better localization is useful but doesn’t fundamentally provide a new way of pretraining. Moreover the improvements in localization are paired with a slight drop in classification performance.
2. Authors repeatedly say their framework “generalized beyond occlusions and deformations”, however the experiments or introduction doesn’t clearly describe what this means.
3. One good clarification would be to validate if DropPos and MAE used any augmentations like translation and zooming, or even using a dynamic grid size for training. All of these could be seen as going beyond a fixed grid sampling.
4. Comparison of the grid size between MAE/DropPos vs PART. Authors sampled 2048 patches, how different is it from other methods?

**Justification:**

1. The main insights seem to be more on optimization of existing methods and do not improve our understanding fundamentally or provide new methods for self-supervised learning.
2. Existing work such as LIO [69] or other methods in related work seem to provide similar insights and therefore further reduce the novelty in this work.
3. There are specific methods that focus on improving object detection using SSL, such as HASSOD: Hierarchical Adaptive Self-Supervised Object Detection. Please provide comparison with these methods (along with more recent approaches) as well since the proposed method focuses more on localization tasks.

---

> ### Author Rebuttal · Authors · 2025-10-22
>
> We thank the reviewer for their thoughtful feedback and for acknowledging the clarity of our methodology, the informative ablations, and the extension of PART to time series data. We address the raised points below.
>
> ### 1. Conceptual contribution and performance
> We would like to clarify that PART is not an optimization of grid-based masked pretraining to off-grid sampling, but operates in the continuous coordinate space of the entire image plane, i.e., at the pixel level. Our goal is to move from patch to pixel level, and our focus is on localized, precise tasks rather than global classification tasks. PART is still competitive on global classification tasks, despite it being the intention of the paper, and achieves higher performance in precise spatial understanding tasks. This trade-off is expected because PART explicitly emphasizes local geometric reasoning.
> This approach enables geometry-aware feature learning, which we show generalizes across modalities (2D images and 1D EEG). Plus, the qualitative capabilities—off-grid reconstruction, symmetry, uncertainty estimation, and scale/aspect-ratio extension—demonstrate properties not captured by grid-based methods. We hope this makes our approach and focus clearer.
> ### 2. Generalizing beyond occlusions and deformations
> Our intention regarding occlusions was to describe how PART learns relative composition between object parts that remain consistent even under occlusion. Similarly, in a deformed or stylized setting—such as in Picasso’s abstract portraits—although the parts may differ in scale, shape, or style, their relative positioning remains coherent in most cases. Since our point on occlusions and deformations is anecdotal, we have changed the tone of the paper to reflect this.
> ### 3. Augmentations and off-grid sampling
> MAE and DropPos were not trained with augmentations such as translation, zooming, or dynamic grid sizes during pretraining; hence, they always operate on the grid. We will make this clear in the paper.
> ### 4. Patch size, number of patches, and number of patch-pairs
> The patch size and number of patches in PART are the same as in MAE and DropPos, as they depend solely on the backbone configuration. For example, with a ViT-B/16 backbone, the patch size is 16×16 pixels, and the number of patches per image is $(224 / 16)^2 = 196$. Unlike MAE and DropPos, which operate on individual patches, PART operates on patch-pairs. Throughout this work, we use a fixed set of 2048 patch-pairs in the relative encoder.
> ### 5. Related work
> In LIO [69], the model learns the structure of an object for each label by first localizing its most prominent part and then modeling the relative positions of other parts with respect to it. Similarly, HASSOD identifies objects in an image through clustering and progressively refines object understanding hierarchically by discovering object parts. Both approaches are tailored to object-centric tasks. In contrast, PART is applicable to both object-centric and object-agnostic data, such as EEG or satellite imagery. We will expand the related work section to position PART more clearly among methods that model part-to-whole compositions.

---

### Official Review · Reviewer_KDLR · 2025-10-10

**Rating:** 4
**Confidence:** 4

**Summary:**

This paper proposes PART (PAirwise Relative Translations), a self-supervised pretraining framework that learns representations by predicting continuous relative translations between random off-grid image patches, rather than predicting absolute grid positions as in MAE, MP3, or DropPos. The key idea is that modeling relative spatial relationships captures the compositional structure of images and generalizes better to occlusions, deformations, and non-rigid object layouts. Additional qualitative studies demonstrate PART’s abilities in off-grid reconstruction, uncertainty estimation, negative symmetry learning, and aspect-ratio-aware sampling. The authors argue that PART could serve as a foundation for universal pretraining across 1D–2D–temporal modalities.

**Strengths:**

1. The paper clearly identifies a limitation of grid-based masked pretraining (rigidity and reliance on absolute positions) and proposes a clean, elegant solution through continuous relative translations.

2. Figures 1–6 are intuitive and effectively communicate the distinction between PART and conventional MIM methods.

3. The ablations (sampling strategy, encoder type, number of patch pairs) and qualitative studies (symmetry, uncertainty) provide convincing insight into why the method works.

**Weaknesses:**

1. While PART is elegant, improvements over strong baselines (e.g., MAE + DropPos) are modest (~+0.3–0.5 AP on COCO). The paper’s conceptual novelty may not be fully matched by substantial downstream benefits.

2. The effect of #pairs, patch size, and computational overhead is only partially discussed. A fair wall-clock or FLOP comparison with MAE would strengthen the claim of efficiency.

3. Some sections (especially 4.2–5.1) are verbose and could benefit from clearer hierarchical organization and concise mathematical notation.

**Justification:**

I recommend Accept for NLDL 2026.
PART provides a conceptually novel and experimentally validated contribution to self-supervised learning, showing promise for flexible, geometry-aware pretraining beyond rigid grid structures. While the performance improvements are moderate and the theoretical depth could be expanded, the work’s clarity, completeness, and potential influence on future multimodal SSL research justify acceptance.

---

> ### Author Rebuttal · Authors · 2025-10-22
>
> We thank the reviewer for their feedback and positive words on the conceptual novelty, clarity, and potential for future multimodal SSL research. Below, we address the raised points.
>
> ### 1. Magnitude of performance improvements
> The numerical improvements are dependent on the task, where we find the biggest improvements for fine-grained and localized tasks. Specifically, +3.4 Cohen’s Kappa over DropPos and +2.3 AP over MAE, +0.6 AP over MP3, and +0.3 AP over DropPos on COCO detection. PART achieves this performance without relying on positional embeddings or auxiliary losses (e.g., position smoothing or attentive reconstruction) used by the baselines. Plus, PART offers additional insights and capabilities unavailable in baseline models, including off-grid reconstruction, symmetry learning, uncertainty estimation, and extension to arbitrary scales and aspect ratios, as well as applicability to both 1D and 2D data. We agree that on coarse-grained and global tasks, our approach obtains more modest results; hence, we focus our claims on fine-grained and localized tasks.
> ### 2. Efficiency
> We apologize for the confusion. When discussing efficiency, we meant sample efficiency. To evaluate sample efficiency, we varied the number of subjects used for fine-tuning in the EEG sleep staging task from 10 to 657, repeated 5 times with different random seeds and sample subsets. As shown below, PART consistently outperforms baselines across all label fractions with fewer fine-tuning samples. PART slightly increases the wall-clock time by 17% compared to DropPos. We will clarify this in the paper and include the additional insights regarding sample efficiency and runtime performance in the paper.
>
> | # Fine-tuning Samples | MP3 | DropPos | MAE | PART |
> |:----------------------:|:---:|:--------:|:---:|:----:|
> | 10  | 0.56 | 0.58 | 0.62 | **0.64** |
> | 50  | 0.63 | 0.64 | 0.65 | **0.67** |
> | 100 | 0.65 | 0.66 | 0.67 | **0.68** |
> | 657 | 0.68 | 0.68 | 0.68 | **0.70** |
>
> ### 3. Verbosity and clarity
> Thank you for the guidance. We have improved writing by reorganizing Section 4 into clearer subsections (“Sampling,” “Objective,” “Architecture,” and “Training Setup”), streamlining mathematical notations, adding a bridging paragraph before Section 5, and subdividing Section 5 into (a)–(d) with short intros.

---

### Meta-Review · Area_Chair_4Tu2 · 2025-10-31

**Recommendation:** Accept (Poster)
**Confidence:** 4

**Metareview:**

This work presentss PART, a self-supervised method that learns representations by predicting relative translations between image patches. Initial reviews were positive, praising the novel and well-motivated approach, clear presentation, and for the most part strong empirical results. Key concerns centred on the modest classification gains, the lack of a detailed computational analysis. The authors rebuttal effectively addressed these points by clarifying the method's focus on localisation, providing a computational overhead analysis (≈15–20% increase vsDropPos), and demonstrating better sample efficiency. Following the rebuttal, reviewers confirmed their concerns were assuaged and maintained their positive scores. The consensus is that the paper presents a technically sound and valuable contribution to self-supervised learning, and the AC agrees with this assessment, recommending the paper for acceptance as a poster presentation.

---

### Decision · Program_Chairs · 2025-11-05

**Decision:**

Accept (Spotlight)

**Comment:**

We recommend an oral and a poster presentation given the AC and reviewers recommendations.

A spotlight presentation refers to a poster selected for an oral highlight but not designated as a full oral presentation per the AC’s recommendation.